# Sulfur Starvation, Sulfide Supplementation, and *cysM* Transcription in *Campylobacter jejuni* Strains with a Single Nucleotide Polymorphism

**DOI:** 10.3390/microorganisms14010097

**Published:** 2026-01-01

**Authors:** Nereus W. Gunther, Aisha Abdul-Wakeel, Manita Guragain

**Affiliations:** Characterization and Interventions for Foodborne Pathogens Research Unit, Eastern Regional Research Center, Agricultural Research Service, United States Department of Agriculture, Wyndmoor, PA 19038, USA; aisha.abdulwakeel@usda.gov (A.A.-W.); manita.guragain@usda.gov (M.G.)

**Keywords:** *Campylobacter jejuni*, sulfur, transcription, *cysM*

## Abstract

The amino acid cysteine is essential to *Campylobacter jejuni* survival, providing the bacterial cells with the element sulfur. When cysteine is not available for uptake, *C. jejuni* can synthesize cysteine from serine and sulfide or thiosulfate. The *cysM* gene produces a cysteine synthase protein required for this process. Transcriptional control for *cysM* has been shown to reside within an untranslated sequence directly upstream of the gene. The untranslated sequence contains a conserved single-nucleotide polymorphism that was previously shown to influence gene transcription. Identification of the 5′ end of the *cysM* mRNA transcript confirmed that the SNP is present within full-length gene transcripts. A new sulfur starvation medium was deployed to study the survival and *cysM* transcription of *C. jejuni* strains with different SNP forms in the presence or absence of sulfide. Differences in survival between the SNP forms were observed during supplementation with low concentrations of sulfide. Additionally, differences in the *cysM* transcription profiles between the same strains with different SNP forms were observed when supplemented with a range of sulfide concentrations. The results support the hypothesis that a gene regulatory element is localized to the area around the SNP in the untranslated region upstream of *cysM*.

## 1. Introduction

*Campylobacter jejuni* remains a highly problematic human pathogen, responsible for the greatest number of bacterial foodborne gastrointestinal disease cases yearly in the developed world [1,2]. In the United States alone, it is estimated that 1.5 million cases of disease are caused annually by Campylobacter, resulting in a financial burden of USD 1.7 billion [3]. In the European Union, the economic loss caused by Campylobacter is estimated at EUR 2.4 billion annually [4].

Poultry products have been implicated as a significant vector for the introduction of *C. jejuni* into humans [2,5,6,7]. In order to persist within poultry products, processing plants, food storage, and preparation conditions to cause the observed disease numbers, *C. jejuni* must be able to survive a series of different environments and overcome multiple environmental stresses [8,9,10]. In addition to needing to overcome these significant challenges, *C. jejuni* requires a microaerobic environment for growth and can be considered nutritionally fastidious [11,12,13,14,15,16]. Among the nutritional requirements of *C. jejuni* is the amino acid cysteine, which is essential to the bacteria’s survival, providing sulfur for the Fe-S clusters found in proteins [17,18].

When *C. jejuni* is unable to acquire cysteine from the environment, the bacterium is able to synthesize it from the amino acid serine and a sulfur source in the form of sulfide or thiosulfate [15,17,19]. While the biochemistry of cysteine synthesis has been well studied, the mechanisms for controlling the genes required for the process need additional research. The final *C. jejuni* gene in the cysteine synthesis process, *cysM*, was initially characterized several years ago [20]. More recently, a single-nucleotide polymorphism (SNP) was identified in the untranslated region (UTR) directly upstream of the *cysM* gene, which previously was shown to contain the gene transcription promoter sequence [21,22]. Additionally, the SNP in the UTR was observed to influence the transcription of *cysM* as well as an exogeneous “reporter” gene (*flaA*) cloned downstream of the sequence [22]. The SNP’s effect on transcription is believed to be at least in part caused by a reduction in full-length transcripts being produced from the genes.

The following research is required to determine how the SNP contributes to the regulation of *cysM* transcription and how that regulation influences *C. jejuni*’s respond to changes in the amounts of environmentally available cysteine: The transcription start site for *cysM* will need to be experimentally identified in order to determine if and where the SNP is located in the gene transcript. Additionally, a medium without cysteine or any source of sulfur that only supports *C. jejuni* survival when supplemented with sulfide will be needed to develop the experimental conditions to study *cysM* transcription. Learning how to manipulate and exploit the control mechanisms of this essential gene will help advance research into developing inhibitors to block *C. jejuni*’s and other related bacteria’s ability to scavenge sulfur and produce cysteine. Recent research on Salmonella has looked at peptides based on the bacteria’s regulatory proteins and metabolites derived from fungus that have demonstrated success inhibiting *cysM* [23,24,25,26].

## 2. Materials and Methods

**Bacterial strains and growth conditions:** All *C. jejuni* strains utilized in this research study are listed in Table 1. Strains were maintained as frozen stocks in Brucella broth (Becton Dickinson, Sparks, NV, USA) containing 15% glycerol at −80 °C. When strains were removed from frozen storage, they were placed directly onto Brucella agar (1.5%) plates. For bacterial growth and during all presented experiments, *C. jejuni* strains were incubated within a microaerobic (5% O_2_, 10% CO_2_, and 85% N_2_) growth chamber (Concept-M, Baker, Sanford, FL, USA) at 42 °C.

**5′ RACE study:** Transcription start points for *C. jejuni*’s native *cysM* gene and for a cloned *flaA* gene fused to the untranslated sequence, taken from the sequence directly upstream of the *cysM* gene, were determined using Invitrogen’s 5′ RACE system for Rapid Amplification of cDNA Ends, version 2.0 (ThermoFisher Scientific, Waltham, MA, USA), following the procedure recommended by the manufacturer. Total RNA was isolated from strain RM3194Δ*flaA*B::tet +*flaA*comp (A20), with primer *cysM_cDNA1* used to synthesize first-strand cDNA from the native *cysM* mRNA and primer *flaA_cDNA1* to synthesize first-strand cDNA from the mRNA produced by the cloned *flaA* gene fused to the UTR. The two cDNA samples were then purified using S.N.A.P. columns included with the RACE kit, and the purified cDNA samples were each subjected to a TdT-tailing reaction. The tailed cDNA samples were amplified using the kit-provided Abridged Anchor Primer in conjunction with the primer cysM5_R for the native *cysM* cDNA or primer *flaA*_qPCR2B_R for the cloned *flaA* cDNA. Next, the resulting PCR products were visualized and separated using agarose gel electrophoresis. Gel bands close to the expected PCR product size were purified from the gel using the QIAquick Gel Extraction Kit (Qiagen, Hilden, Germany). The purified products were cloned into the pCR™4-TOPO™ TA vector (ThermoFisher Scientific) and amplified within *E. coli* TOP10 cells. Finally, the plasmids containing the cloned products were sequenced to determine the transcription start sites.

**Sulfur starvation medium and sulfur supplementation survival:** A *C. jejuni*-specific medium devoid of any essential sulfur source was developed using the efforts of previous labs as a starting point [17,19]. The medium was designed to be, when supplemented with sulfide, sufficient to support the survival of *C. jejuni* cells but not sufficient for expansion of the bacterial population (Table 2). This new sulfur starvation medium (SSM) was used for a series of *C. jejuni* survival experiments where the starvation medium was supplemented with a range of different concentrations (stock concentrations: 0 mM, 20 mM, 100 mM, and 200 mM) of anhydrous sodium sulfide, technical grade (PPG Industries, Inc., Pittsburg, PA, USA). Strains of *C. jejuni* used in the sulfur supplementation survival experiments were grown overnight on Brucella agar plates and then used to inoculate Brucella broth cultures that were incubated for sixteen hours in microaerobic conditions at 42 °C. The resulting bacterial cultures were centrifuged for 5 min at 7000× *g*, forming a bacterial pellet. The pellet was washed once with 1 mL of SSM and then pelleted again before being resuspended in 100 µL of fresh SSM. The cells were then diluted to a starting concentration of between 108 and 109 cells per mL into fresh SSM containing the desired amount of sodium sulfide for the specific experiment. Subsequent survival of *C. jejuni* strains in these experiments was measured by determining the colony-forming units present within individual samples using the 6 × 6 drop plate method [28].

**Transcription analysis of** ***flaA*** **and** ***cysM*****:** For gene transcriptional studies presented in this manuscript, RNA was initially isolated from cells pelleted from 2 mL of bacterial samples, grown under various defined conditions, using the Invitrogen PureLink RNA Kit (ThermoFisher Scientific). The resulting whole RNA samples were treated with Invitrogen’s Turbo Dnase kit (ThermoFisher Scientific) to remove any contaminating DNA. The DNased RNA samples were then used to produce cDNA utilizing the SuperScript IV VILO Master Mix kit (ThermoFisher Scientific).

Subsequent quantitative PCR (RT-PCR) studies to analyze the relative cDNA amounts from *C. jejuni* samples were conducted using a Roche LightCycler 96 system (Roche Scientific, Branchburg, NJ, USA). The housekeeping gene rpoA was utilized as an internal control allowing for the normalization of different starting RNA levels between samples that were to be compared [29]. Primers *cysM*2G_F and *cysM*2G_R were used to measure native *cysM* transcription, and primers *flaA*2E_F and *flaA*2E_R were used to measure transcription of the cloned *flaA*. The resulting quantitative PCR data were used to calculate the relative transcription level of the target gene (*cysM* or *flaA*) as a ratio to the housekeeping gene (rpoA) to allow for comparisons between strains and conditions. Equation: ratio = ERCqR/ETCqT (ER = quantification efficiency of reference gene, ET = quantification efficiency of target gene, CqR = quantification cycle of reference, and CqT = quantification cycle of target.). The RT-PCR reactions were performed in white LightCycler 96-well plates (Roche) in 20 µL total reaction volumes (10 μL of FastStart Essential DNA Green Master Mix (Roche), 1 μL of each 10 μM primer forward and reverse, and 8 μL of cDNA previously diluted 1:40 in water). The RT-PCR protocol was as follows: initial denaturation step at 95 °C for 10 min, followed by 45 cycles of 95 °C for 10 s, 58 °C for 10 s, and 72 °C for 10 s, concluding with a melting curve analysis. Each RT-PCR experiment included “no reverse transcriptase” samples to control for genomic DNA contamination in the cDNA preparations.

**Statistical analysis:** All mean and standard error values reported in this study are the result of at least three separate experimental replicates, with each experimental replicate having multiple technical replicates. One-way analysis of variance (ANOVA) tests were used to compare mean values within experiments and to determine if significant differences between the mean values exist. Following any ANOVA tests, a Tukey HSD analysis was used to separate the mean values. For the results presented in this manuscript, significant differences were defined as *p*-values ≤ 0.05.

## 3. Results

### 3.1. Transcriptional Start Site of cysM

The promoter control for the *cysM* gene of *C. jejuni* was localized to the 126-base-pair untranslated region (UTR) directly upstream of the gene [20,21,22]. However, the transcriptional start site (TSS) for *cysM* within the UTR has not been experimentally determined. To accomplish this goal, the *C. jejuni* strain RM3194Δ*flaA*B::tet +*flaA*comp (A20) and a 5′ RACE technique were utilized to determine the TSS. Strain RM3194Δ*flaA*B::tet +*flaA*comp (A20) was chosen for this study because it possesses the native UTR and *cysM* sequences and a cloned UTR fused to a cloned *flaA* gene within the bacteria’s genome [22]. Determining the TSS for both UTR and gene combinations will determine if control of the TSS is specific to the UTR sequence or if it is influenced by sequences outside of the UTR.

For the UTR-*cysM* combination, twenty unique transcripts were sequenced, and for the UTR-*flaA* combination, ten unique transcripts were sequenced to determine the TSS. The greatest number of transcripts terminating at the same sequence location (TTTTGA) were observed to occur within the UTR, thirty-eight nucleotides upstream of the gene translation start codon (ATG) (Figure 1). This was observed for both UTR gene combinations, with four transcripts terminating at this site for the UTR-*cysM* combination and five transcripts for the UTR-*flaA* combination. The UTR-*cysM* combination also had one transcript that terminated slightly downstream of the others at a location twenty-five base-pairs upstream of the translational start. It was interesting to observe that both UTR gene combinations had a significant number of transcripts with transcription start sites within the coding region of the *cysM* or *flaA* gene, a result that would theoretically produce a truncated protein product. The UTR-*cysM* combination had fifteen transcripts with transcriptional start sites internal to the *cysM* gene, representing 75% of the transcripts sequenced, while the UTR-*flaA* combination had five transcripts with five gene internal transcription start sites, comprising 50% of the transcripts sequenced. All transcripts with transcriptional start sites within the gene were found to occur within the first two hundred nucleotides of the gene’s coding region. Finally, it is important to note that the transcriptional start site within the UTR promoter region would produce transcripts that would include the previously observed single-nucleotide polymorphism (SNP) region of the UTR. This result would be consistent with the previously published study that hypothesized that the SNP site is located in a position to influence gene transcription [22].

### 3.2. Campylobacter jejuni Survival During Sulfur Starvation and Sulfide Supplementation

Previous research investigating *cysM* function and *C. jejuni* sulfur dependance developed a minimal medium that, when supplemented with inorganic sulfur, appeared to allow bacterial growth [19]. In order to further study the effects of sulfur starvation on *C. jejuni*, defined media from the previous research were used as a starting point to develop a simpler sulfur starvation medium (SSM). To produce the SSM, any material in the previous medium that was not essential was removed, resulting in a medium with a minimal number of components that was devoid of any sulfur (Table 2). The SSM, when supplemented with a sulfur source, was designed to only support *C. jejuni* survival and not the expansion of the bacterial population. Strains RM3194 and CDC9511were selected for experiments using the SSM, since they differed with regard to the SNP site in the transcribed region of the UTR upstream of *cysM* (RM3194 has an adenine at the SNP site, while CDC9511 has a guanine at the same site).

Three milliliter cultures of RM3194 were incubated microaerobically at 42 °C in SSM supplemented with a range of sodium sulfide solution concentrations (0 mM, 20 mM, 100 mM, and 200 mM). The sodium sulfide solutions were added as 10 µL volumes every hour of incubation, with 10 µL of water being added as the 0 mM concentration. Samples were collected at 0, 3, 6, 9, 12, and 16 h, and the viable cell count per ml for the different cultures was determined (Figure 2A). Within three hours after the initial inoculation into the SSM, the samples without supplemental sulfide showed a significant decrease compared to all samples with added sulfide. At all subsequent timepoints measured, the samples without added sulfide continued to demonstrate a significant decrease in viable cells compared to the samples with added sulfur, regardless of the concentration of the stocks used. While significant, the overall decrease in cells in the samples without added sodium sulfide was only just over one log in total cell number reduction during the length of the experiment (16 h). Also, in terms of viable cell numbers, there was little change between samples with added sodium sulfide from different stock concentrations during the length of the sixteen-hour experiment.

When the same experiment was repeated using strain CDC9511, the results were similar, with notable differences (Figure 2B). There was a significant decrease in all samples regardless of supplemental sodium sulfide concentrations during the initial eight hours of the experiment. At twelve hours, the cell numbers of the samples supplemented with sodium sulfide stocks of 100 mM and 200 mM concentrations recovered to the level observed at the zero timepoint, while the numbers in the samples supplemented with 20 mM or 0 mM stocks continued to decrease. The significant differences in cell survival numbers continued at the twelve- and sixteen-hour sampling points between samples supplemented with 100 mM and 200 mM sodium sulfide stocks and those supplemented with 20 mM and 0 mM. At the sixteen-hour timepoint, the differences in the mean values between these two groups were over two logs in bacterial survival numbers.

The previous experiments followed sulfur starvation and supplementation over sixteen hours but only observed overall modest reductions (1–2 logs) in cell numbers. Parts of the previous experiments were extended to observe sulfur starvation in strain RM3194 over a longer time period of forty hours. Initially, the results were similar to those previously observed with RM3194 (Figure 2C). After sixteen hours, roughly a one-log difference in viable cells was again observed for samples supplemented from a 100 mM stock of sodium sulfide compared to samples not supplemented. As the experiment continued, the difference in viable cells increased to over two logs at twenty-four hours to over five logs when the experiment was discontinued at forty hours. In this extended, experiment sodium sulfide supplementation occurred every fifteen hours versus every hour as was done in the previous experiments. This is likely to account for the observed up and down variation in the cell viability of the sodium sulfide-supplemented samples that was not observed in the previous results (Figure 2A).

### 3.3. Gene Transcription Dynamics During Sulfur Starvation and Supplementation

After determining the survival of *C. jejuni* strains in SSM with and without sodium sulfide supplementation, the influence of the same incubation conditions on the transcription of the *cysM* gene was observed. Previous research had demonstrated that the *cysM* gene is a critical gene for sulfur utilization in *C. jejuni* [19]. The *cysM* gene is responsible for the final step in the process of combining sulfur and serine to produce cysteine, an amino acid essential to *C. jejuni*. Since the SSM is devoid of cysteine, in previous survival experiments, the *cysM* gene had to produce cysteine from the serine in the media and the supplemented sodium sulfide in order to keep the *C. jejuni* cells alive. Quantitative RT-PCR was performed to observe how different concentrations of sodium sulfide might change *cysM* transcription levels and how the two different SNP forms of the *cysM* UTR might further influence this transcription.

Samples were collected for RNA isolation from RM3194 incubated for seven hours in SSM supplemented hourly with 10 µL from stocks of sodium sulfide at concentrations of 0 mM, 20 mM, 100 mM, and 200 mM. Seven hours of incubation was selected based on the results of the previous experiment (Figure 2A,B) that showed that significant reductions in RM3194 survival without sodium sulfide supplementation occurred within six hours of the start of incubation in SSM. Significant differences in *cysM* transcription based on the concentration of the sodium sulfide stocks used for supplementation were observed in the samples collected at seven hours post-inoculation into SSM (Figure 3). The RM3194 samples with the least concentrated sodium sulfide solution (20 mM) produced the highest transcription levels in *cysM* compared to those supplemented with the greater solution concentrations (100 mM and 200 mM). The increased level of transcription in the 20 mM compared to the 100 mM and 200 mM sodium sulfide-supplemented samples appears inconsistent with the previously observed cell survival results, where the 20 mM, 100 mM, and 200 mM levels of sodium supplementation resulted in statistically equal levels of survival.

In order to determine if the SNP, in the UTR containing the *cysM* promoter, would have an influence on *cysM* transcription during supplementation with different sodium sulfide concentrations, the previous quantitative RT-PCR experiments were repeated using CDC9511. The observed *cysM* transcription profile for CDC9511 was different from that of RM3194 (Figure 4). For CDC9511, the highest concentration of sodium sulfide supplementation, 200 mM, also produced the greatest amount of *cysM* transcription compared to the 20 mM sodium sulfide concentration. This was the opposite of what was observed for RM3194.

To control for differences between RM3194 and CDC9511, other than the SNP UTR, that might influence *cysM* transcription, two isogenic strains were used to repeat the previous experiments. The strains are genetically identical, except one has the adenine SNP UTR fused to an exogeneous *flaA* gene and cloned into the bacterial chromosome, while the other has the guanine SNP UTR fused to the *flaA* gene. The first of the isogenic strains, RM3194Δ*flaA*B::tet +*flaA*comp (A20), is an RM3194 strain and, therefore, has a native A20 UTR upstream of the *cysM* gene; it also has an additional A20 SNP form of the UTR fused to a *flaA* gene and cloned within the bacterial chromosome. Additionally, the wild-type *flaA* and flaB genes were completely deleted from this bacterial strain. The transcription profile for the *cysM* gene in RM3194Δ*flaA*B::tet +*flaA*comp (A20) supplemented with different concentrations of sodium sulfide was similar to that previously seen with the RM3194 parent strain (Figure 5A). The lower concentration of sodium sulfide supplementation, 20 mM, produced higher *cysM* transcription levels compared to the samples treated with greater concentrations, 100 mM and 200 mM, of sodium sulfide. This same transcription profile was observed for the *flaA* gene fused to the A20 SNP form of the UTR (Figure 5B). The 20 mM sodium sulfide supplementation again produced greater *cysM* transcription levels compared to the samples treated with 100 mM and 200 mM stocks of sodium sulfide.

Next, the second isogenic strain, RM3194Δ*flaA*B::tet +*flaA*comp (G20), with the G20 form of the UTR in front of the cloned *flaA* was used to repeat the previous experiments. Strain RM3194Δ*flaA*B::tet +*flaA*comp (G20) is an RM3194 strain with a native A20 UTR upstream of the *cysM* gene but with the G20 SNP form of the UTR fused to a *flaA* gene and cloned within the bacterial chromosome. Not surprisingly, the native A20 UTR-*cysM* of RM3194Δ*flaA*B::tet +*flaA*comp (G20) had a transcription profile similar to that of RM3194 and RM3194Δ*flaA*B::tet +*flaA*comp (A20), with the lesser concentration of sodium sulfide, 20 mM, producing greater *cysM* transcription compared to the higher concentration of 200 mM (Figure 6A). However, the cloned G20 UTR and *flaA* combination in RM3194Δ*flaA*B::tet +*flaA*comp (G20) had a transcription profile similar to A20 UTR-*flaA* of RM3194Δ*flaA*B::tet +*flaA*comp (A20) and not like the native G20 UTR-*cysM* of strain CDC9511 (Figure 6B), which was contrary to expectations.

## 4. Discussion

In the presented study, the experimentally determined TSS for the *cysM* gene is consistent with data from previous *C. jejuni* whole-transcriptome mapping [30]. In addition to determining the TSS, three additional observations can be made from the sequences of DNA produced from the *cysM* transcripts. This first observation is that the UTR region sequence alone is sufficient to determine the TSS and is not dependent on the gene sequence of *cysM*, since it was possible to fuse the UTR to an exogenous *flaA* sequence and produce the same TSS for this combination (Figure 1B). The second observation is that the TSS produces transcripts that include a previously described SNP region (Figure 1) [22]. The SNP had been previously implicated in producing varying effects on gene transcription possibly through a blocking of proper transcription from the TSS. The observations that the SNP was present at the 5′ end of full-length *cysM* mRNA transcripts and that under certain conditions the 5′ end of mRNA transcripts originating from the UTR were truncated [22] supports the theory that the SNP could play a role in blocking the full-length transcription required for successful cysteine synthase production. Another relevant observation is that a large percentage of the sequenced transcripts produced for this study had apparently random TSSs within the first two hundred nucleotides of the translated portion of the *cysM* gene. Again, this supports the theory that the SNP region is in a position to block transcription from the proper TSS, leading to random transcription starts within the gene coding region, which would then be expected to produce truncated non-functioning proteins. The blocking of proper transcription must be caused, at least in part, by the UTR, since the transcription starts within the gene occurred with the native *cysM* as well as when the UTR is cloned in front of an exogeneous copy of *flaA*, suggesting that the gene sequence does not play a role in the random transcriptional start sites (Figure 1B). The exact nature and function of this gene control mechanism could prove valuable given that when cysteine is limited in the environment, *cysM* is the gene responsible for the production of this essential amino acid from inorganic sulfur sources [19,20].

To study the transcription profiles of *C. jejuni* during sulfur starvation, it was determined that a new SSM would have to be developed. The previous medium for studying sulfur usage in cysteine biosynthesis was rejected because it appeared to facilitate bacteria growth when supplemented with sulfur as opposed to only supporting cell survival, as was desired for the presented studies [17,19]. Additionally, the previous media contained methionine and other sulfur sources, which, during initial experiments, appeared to support *C. jejuni* survival. It should also be noted here that the presented research used direct plating and CFUs to follow cell survival, as opposed to optical density measurements, which we have found can be misleading given *C. jejuni*’s ability to form non-culturable/non-viable coccoid forms [31]. The SSM designed for the current experiments fulfills the requirement to sustain *C. jejuni* survival for an extended period of time (~40 h) when supplemented regularly with sodium sulfide (Figure 2C). Additionally, the SSM without sulfur produces a steady decline in cell viability over the same time period. Use of SSM and sulfide supplementation produced data that suggested a logical point to collect samples for transcriptional studies as well as differences in survival between *C. jejuni* strains when exposed to different sodium sulfide concentrations (Figure 2A,B). The sulfide-mediated differences between RM3194 and CDC9511 survival in SSM could be influenced by the strains’ UTR SNPs. Therefore, determining the transcription profiles for these strains had the potential for contributing to understanding how the *cysM* gene is regulated in *C. jejuni*.

The transcription profile of RM3194 in SSM supplemented with a range of different sodium sulfide concentrations was not as initially expected. Since sodium sulfide was required for survival, it was assumed that the higher the concentrations of sodium sulfide, the greater the levels of transcription; however, we observed the opposite (Figure 3). The smaller concentrations of sulfide available in the environment led to increased *cysM* transcription, suggesting that some sort of feedback mechanism may exist in *C. jejuni* that reduces *cysM* transcription when sulfur is plentiful. This observation was reinforced by the transcription profile of RM3194Δ*flaA*B::tet +*flaA*comp (A20), which shows the same increased transcription at lower sulfur concentrations for an exogeneous *flaA* gene when it is fused to the *cysM* UTR (Figure 5). It is important to remember that RM3194 and RM3194Δ*flaA*B::tet +*flaA*comp (A20) have the adenine SNP version of the UTR in front of *cysM* or the cloned *flaA*. Strain CDC9511, on the other hand, has the guanine SNP version of the UTR in front of *cysM* and produced a transcription profile where the maximum transcription results from the higher concentration of sulfide, the opposite of the result from the previous two strains (Figure 4). This suggests that if there is a feedback mechanism reducing *cysM* transcription when sulfide is more available in the environment or enhancing *cysM* transcription when the cells are starved for sulfide, then the theoretical mechanism depends on the adenine being present in the SNP of the UTR, and the guanine nucleotide disrupts this level of gene control. In *E. coli* and *S. typhimurium*, sulfide reduces the transcription of the *cysM* orthologues by acting as an anti-inducer of the regulatory gene cysB, which is responsible for activating the transcription of the *cysM* in these bacteria [32,33,34]. However, *C. jejuni* lacks an orthologue of cysB; therefore, if sulfide at certain concentrations is acting to reduce *cysM* transcription, it must be through an alternate mechanism [19]. Sulfide acting as an inhibitor of *cysM* might also help explain the observed differences in survival between RM3194 and CDC9511 incubated in SSM and supplemented with 20 mM sodium sulfide (Figure 2A,B). Since CDC9511 shows significantly reduced survival at the 20 mM level of sodium sulfide supplementation compared to RM3194, it is possible to imagine that the guanine SNP reduces *cysM* transcription at a level that does not allow for sufficient production of essential cysteine from the sulfide present in the environment. Therefore, CDC9511 may need significantly higher concentrations of sodium sulfide present to overcome the blocking of *cysM* transcription caused by the guanine SNP. Unfortunately, the transcription profiles of strain RM3194Δ*flaA*B::tet +*flaA*comp (G20) do not provide support for the previous hypothesis (Figure 6A,B). Strain RM3194Δ*flaA*B::tet +*flaA*comp (G20) has a native adenine SNP UTR driving *cysM* transcription, which produces a transcription profile similar to all other adenine SNP possessing UTRs observed. However, the strain also has a guanine SNP UTR driving the cloned *flaA* transcription, and it would be expected that this would provide a transcription profile similar to the *cysM* profile of CDC9511. Instead, the guanine SNP UTR and *flaA* combination has a transcription profile similar to the adenine SNP UTR and *flaA* combination. These conflicting data will need to be sorted out by an expanded transcriptional study, which should include RNA-Seq comparisons of all RNA transcripts of RM3194, CDC9511, RM3194Δ*flaA*B::tet +*flaA*comp (A20), and RM3194Δ*flaA*B::tet +*flaA*comp (G20) incubated in SSM with a range of different sodium sulfide concentrations. This will provide additional data as to whether the adenine and guanine SNP forms have a secondary influence on any other gene transcription, hopefully providing clarity to their primary effects on *cysM* transcription.

*Campylobacter jejuni*’s *cysM* has been identified as an essential gene, making it an attractive target for interventions [35]. The mechanisms controlling transcription of the *cysM* gene have been well studied in *E. coli* and *Salmonella*, bacteria related to *C. jejuni* [36]. However, *C. jejuni*, likely because it possesses a relatively small genome, lacks many of the genes related to sulfur metabolism that are found in other bacteria, including those for control of *cysM* [18,19]. Nevertheless, this might present an opportunity. Given *C. jejuni*’s simpler genome and lack of genes for complex control of the *cysM* gene, it is likely that the transcriptional control of this gene occurs by way of a simpler mechanism. A simpler mechanism may allow for easier understanding and control in order to block transcription of this essential gene. The SNP within *cysM*’s UTR has provided initial data suggesting the sequence in this area is required for sulfide’s repressive influence on the gene’s transcription.

Future experiments will focus on defining the environmental conditions that will maximize *cysM* transcription and determine if those conditions are the same for both identified SNP forms. There will be a secondary focus on identifying an inducing agent that will enhance gene transcription for any gene experimentally fused to the UTR in the same manner that the *flaA* gene was in the presented research. Additionally, whole-genome transcriptional studies will be performed to identify additional genes integral to *C. jejuni* survival during sulfur-limiting conditions.

## Figures and Tables

**Figure 1 microorganisms-14-00097-f001:**
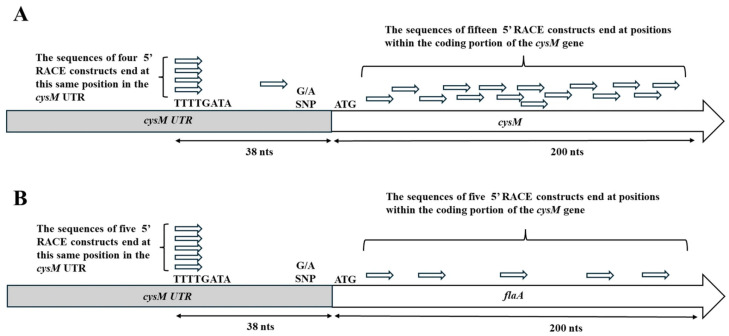
Results of 5′ RACE study of transcripts derived from the native UTR-*cysM* sequence (**A**) and the UTR-*flaA* cloned sequence (**B**). The relative locations of the 5′ ends of the transcripts sequence are displayed in the untranslated region (UTR) of the bacterial genome or within the coding region of the downstream gene sequence. Twenty transcripts were sequenced for the UTR-*cysM* combination and ten transcripts for the UTR-*flaA* combination. The relative position of the previously identified single-nucleotide polymorphism within the UTR was also noted.

**Figure 2 microorganisms-14-00097-f002:**
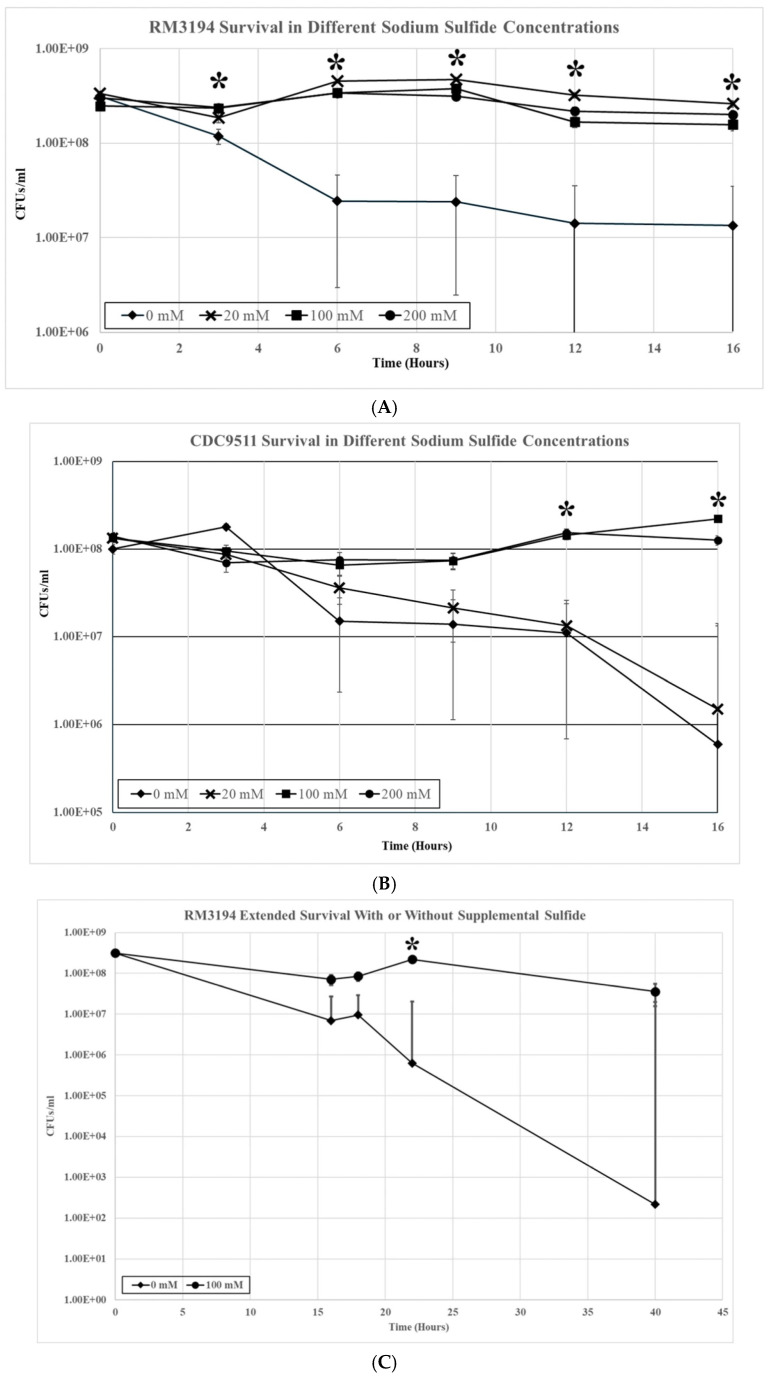
(**A**) RM3194 survival in 3 mL of sulfur starvation medium (SSM) supplemented with different concentrations of sodium sulfide. Sodium sulfide supplementation of 10 µL from solutions with concentrations of 0 mM (water), 20 mM, 100 mM, and 200 mM was performed every hour, and samples to determine the colony-forming units (CFUs) were collected at 0, 3, 6, 9, 12, and 16 h after inoculation into SSM (* = statistical difference between mean CFUs produced by different sample types at specific timepoint). (**B**) CDC9511 survival in 3 mL of sulfur starvation medium (SSM) supplemented with different concentrations of sodium sulfide. Sodium sulfide supplementation of 10 µL from solutions with concentrations of 0 mM (water), 20 mM, 100 mM, and 200 mM was performed every hour, and samples to determine the colony-forming units (CFUs) were collected at 0, 3, 6, 9, 12, and 16 h after inoculation into SSM. (* = statistical difference between mean CFUs produced by different sample types at specific timepoint). (**C**) RM3194 extended survival in 3 mL of sulfur starvation medium (SSM) with or without sodium sulfide. Sodium sulfide supplementation of 10 µL from solutions with concentrations of 0 mM (water) or 100 mM was performed at 0, 16, and 22 h after inoculation into SSM. Samples to determine the colony-forming units (CFUs) were collected at 0, 16, 18, 22, and 40 h after inoculation (* = statistical difference between mean CFUs produced by different sample types at specific timepoint).

**Figure 3 microorganisms-14-00097-f003:**
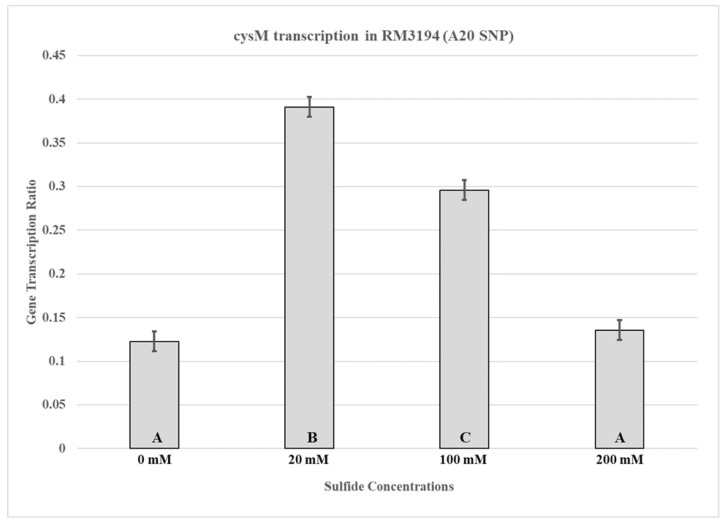
*cysM* transcription in RM3194 (A20 SNP). The strain was incubated in 3 mL of sulfur starvation medium (SSM) supplemented with different concentrations of sodium sulfide for seven hours before cells were collected for RNA isolation. Sodium sulfide supplementation of 10 µL from solutions with concentrations of 0 mM (water), 20 mM, 100 mM, and 200 mM was performed every hour. The mean relative ratio of transcription levels and the resulting standard error values for the data were plotted. Significant differences between the mean transcription values of the samples are indicated by different letter notations.

**Figure 4 microorganisms-14-00097-f004:**
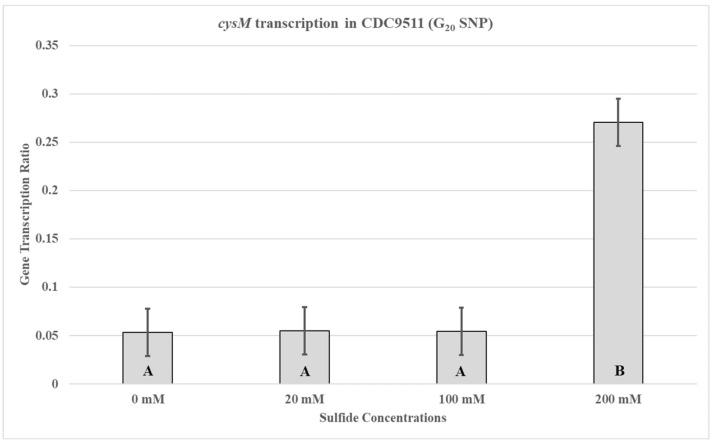
*cysM* transcription in CDC9511 (G20 SNP). The strain was incubated in 3 mL of sulfur starvation medium (SSM) supplemented with different concentrations of sodium sulfide for seven hours before cells were collected for RNA isolation. Sodium sulfide supplementation of 10 µL from solutions with concentrations of 0 mM (water), 20 mM, 100 mM, and 200 mM was performed every hour. The mean relative ratio of transcription levels and the resulting standard error values for the data were plotted. Significant differences between the mean transcription values of the samples are indicated by different letter notations.

**Figure 5 microorganisms-14-00097-f005:**
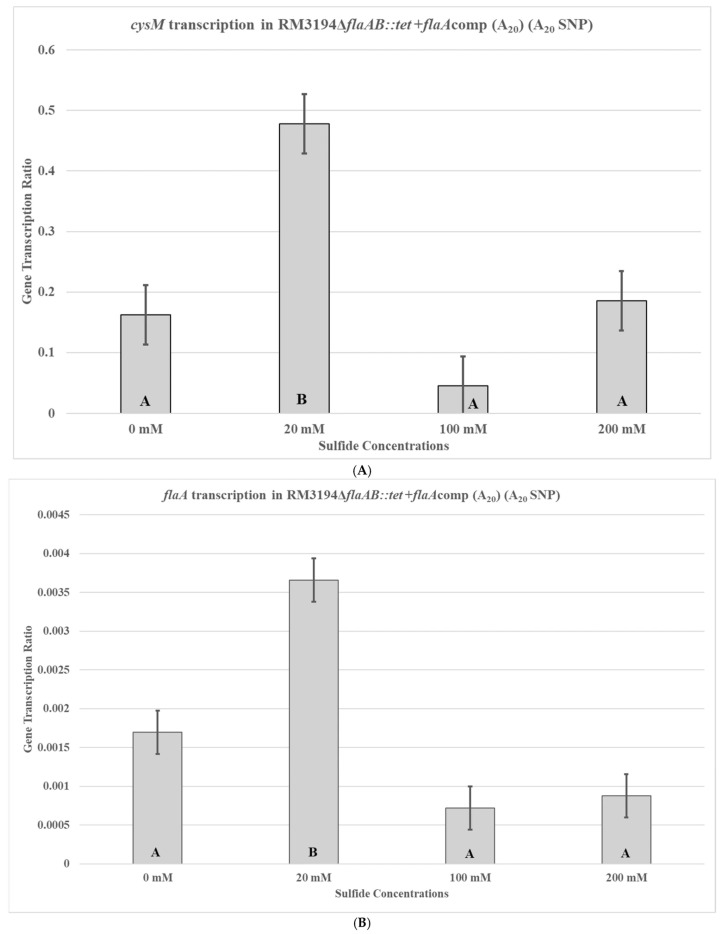
*cysM* (**A**) or *flaA* (**B**) transcription in *C. jejuni* RM3194Δ*flaAB::tet* +*flaA*comp (A_20_). The strain has an A_20_ form of the UTR in front of both the *cysM* and *flaA*. The cloned strain was incubated in 3 mL of sulfur starvation medium (SSM) supplemented with different concentrations of sodium sulfide for seven hours before cells were collected for RNA isolation. Sodium sulfide supplementation of 10 µL from solutions with concentrations of 0 mM (water), 20 mM, 100 mM, and 200 mM was performed every hour. The mean relative ratio of transcription levels and the resulting standard error values for the data were plotted. Significant differences between the mean transcription values of the samples are indicated by different letter notations.

**Figure 6 microorganisms-14-00097-f006:**
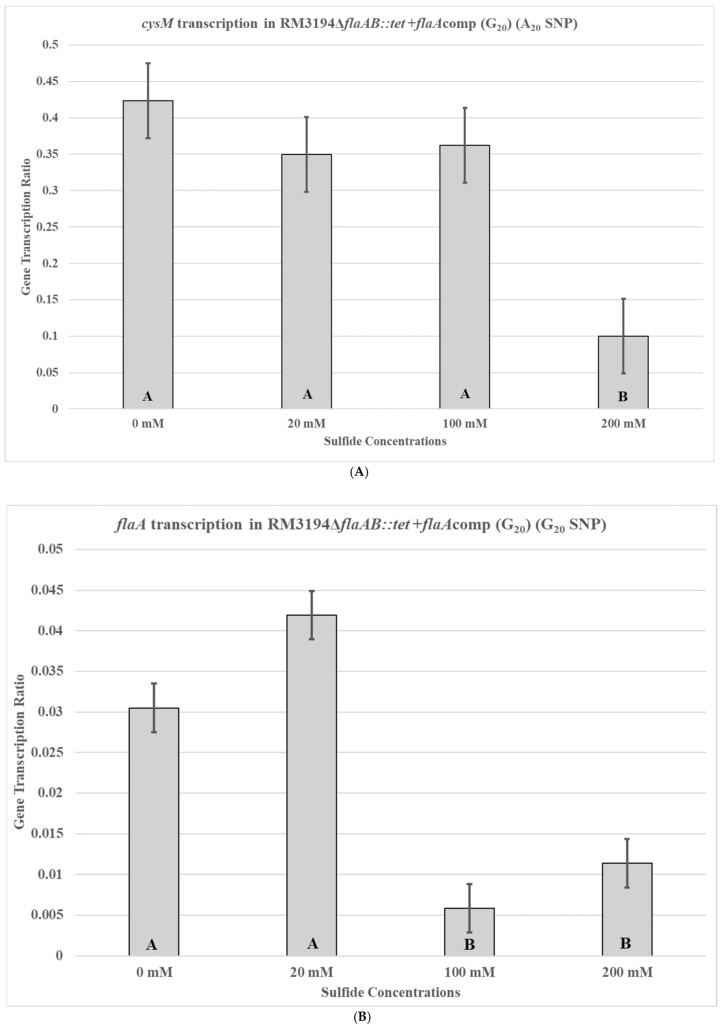
*cysM* (**A**) or *flaA* (**B**) transcription in *C. jejuni* RM3194Δ*flaAB::tet* +*flaA*comp (G_20_). The strain has an A_20_ form of the UTR in front of *cysM* and a G_20_ form of the UTR in front of *flaA*. The cloned strain was incubated in 3 mL of sulfur starvation medium (SSM) supplemented with different concentrations of sodium sulfide for seven hours before cells were collected for RNA isolation. Sodium sulfide supplementation of 10 µL from solutions with concentrations of 0 mM (water), 20 mM, 100 mM, and 200 mM was performed every hour. The mean relative ratio of transcription levels and the resulting standard error values for the data were plotted. Significant differences between the mean transcription values of the samples are indicated by different letter notations.

**Table 1 microorganisms-14-00097-t001:** Strains and primers utilized in research.

Strain	Description	Source or Reference
*C. jejuni* CDC9511	Clinical isolate CDC9511 (2012D-9511)	[22]
*C. jejuni* RM3194	Clinical isolate	[27]
*C. jejuni* RM3194Δ*flaAB::tet* +*flaA*comp (G_20_)	RM3194 with deletion of *flaA* and *flaB*; with integrated *cysM* UTR (G_20_ variety)-*flaA* expression element; Tc^R^ and Kn^R^	[21]
*C. jejuni* RM3194Δ*flaAB::tet* +*flaA*comp (A_20_)	RM3194 with deletion of *flaA* and *flaB*; with integrated *cysM* UTR (A_20_ variety)-*flaA* expression element; Tc^R^ and Kn^R^	[22]
**Primers**	**Description (5′-3′)**	**Source**
rpoA2B_F	tcttcaagcataccacgcat	[22]
rpoA2B_R	atcaccctagcccatccttt	[22]
cysM2G_F	agtatatgaaaaagtaagtgagc	[22]
cysM2G_R	catttcaaaagctgctctatct	[22]
*flaA*2E_F	atgggatttcgtattaacaccaa	[22]
*flaA*2E_R	ctaagacctgaactaagtctgct	[22]
*flaA*_cDNA1	agagccactttgagccaaga	This study
cysM_cDNA1	tgagaggtatctttcagccgt	This study
*flaA*_qPCR2B_R	gaaccaatgtcggctctgat	This study
cysM5_R	gcatacttgcggcaaataca	This study

**Table 2 microorganisms-14-00097-t002:** Sulfur starvation medium (SSM).

Chemical	Final Concentration (mM)
CaCl_2_	1.8
Fe(NO_3_)_3_	0.00025
MgCl_2_	1.75
KCl	5.4
NaHCO_3_	44
NaCl	100
NaH_2_PO_4_	0.9
Niacinamide	0.033
Na-Lactate	10
L-Leucine	0.8
L-Aspartic acid	10
L-serine	20

## Data Availability

The original contributions presented in this study are included in the article. Further inquiries can be directed to the corresponding author.

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
