# Peer review of "Sulfur Starvation, Sulfide Supplementation, and *cysM* Transcription in *Campylobacter jejuni* Strains with a Single Nucleotide Polymorphism"

_microorganisms, 2026, doi:10.3390/microorganisms14010097_

Round 1
Reviewer 1 Report
Comments and Suggestions for Authors
This paper looks into a SNP in the UTR of the gene encoding CysM (cysteine synthase) in C. jejuni. It is nicely written and for the most part well explained and easy to follow.
Mapping the TSS is useful, confirming that the SNP is present within full length transcripts of cysM.
The authors go on to develop a minimal media to support C. jejuni survival/persistence but not active growth, which seems fine.
I got a bit bogged down in last part on gene expression and the different strain combinations used - figs 3 and 4 show different pattern in two strains that differ in the SNP - why not just change the SNP upstream of cysM in each case and see if the pattern is altered rather than the complicated situation with flaA genes?
Throughout the authors say the gene is responsible for cysteine synthesise, when they mean the protein.
Author Response
Reviewer 1: This paper looks into a SNP in the UTR of the gene encoding CysM (cysteine synthase) in C. jejuni. It is nicely written and for the most part well explained and easy to follow.
Author response: Thank you for the generous words.
Reviewer 1: Mapping the TSS is useful, confirming that the SNP is present within full length transcripts of cysM.
Author response: We agree it was important to establish that the SNP would fall within the full length transcripts before further pursuing any investigation of the SNP.
Reviewer 1: The authors go on to develop a minimal media to support C. jejuni survival/persistence but not active growth, which seems fine.
Author response: The new minimal media has been very instructive in our current line of research following the submitted manuscript.
Reviewer 1: I got a bit bogged down in last part on gene expression and the different strain combinations used - figs 3 and 4 show different pattern in two strains that differ in the SNP - why not just change the SNP upstream of cysM in each case and see if the pattern is altered rather than the complicated situation with flaA genes?
Author response: We regret that the experiments for figures 3 and 4 and subsequently figures 5 and 6 can be a bit complicated and agree that the reviewer’s solution of just altering the SNP in a single strain would have been simpler and achieved the same goal. We had a few reasons for the approach we took (listed below), which is not to say we do not intend to make the suggested strain for our continuing studies.
- Strain CDC9511 is not transformable (some jejuni strains are not for currently unknown reasons). However, this could have probably been done with strain RM3194.
- We had previously developed (Uhlich, Biotechniques, 2022) the cloning tools for fusing exogenous genes to the UTR region with either the A or G SNP form. Which made the approach easier and more attractive.
- Using the UTR fused to flaA approach had a couple of advantages. It used a gene (flaA) different than cysM which eliminated the possibility of the specific gene contributing to the transcriptional control that we believed was attributable to the SNP sequence region. Additionally, it moved the entire SNP plus gene sequence to a completely different region in the genome, which suggested that the sequence directly surrounding the UTR and cysM did not play a role in the transcriptional control.
- Finally, given the genetic plasticity of C. jejuni strains we believe it is not best to base our results solely on molecularly manipulated strains that undergo multiple passages during the cloning process. We believe it is beneficial to utilize minimally passaged “wild-type” strains for certain portions of the research. In the past this has provided us with unexpected and useful results.
- However, we do agree with the reviewer’s basic point that a RM3194 strain with a G SNP form to compliment the 3194 strain with the A SNP would be beneficial to our future research. We are currently attempting that construct (it has been challenging).
Reviewer 1: Throughout the authors say the gene is responsible for cysteine synthesise, when they mean the protein.
Author response: We have made changes to the text to properly reflect that the cysM gene produces the cysteine synthase protein and the resulting protein plays a role in the cysteine synthesis process.
Reviewer 2 Report
Comments and Suggestions for Authors
The study presents an important finding for an a serious pathogen, Campylobacter jejuni. Particularly, the cysM transcription finding worths publication and is well explained and supported methodologically. I congratulate the authors for way of presentation and details provided and I have only some minor revisions to propose, as follows:
Please explain better with one additional phrase in the abstract the “Start site”. Not all readers of a microbiological journal are totally familiar with the concept
The first sentence of the last paragraph of the introduction is hard to follow, please rephrase
In discussion, it is referred that “Since the TSS was determined to be slightly upstream of the SNP, it supports the theory that the sequence containing the SNP could play a role in the blocking of full-length transcription.” Why is that? Add a more detailed explanation and define the “slightly”
Finally, I suggest to add a small paragraph in the discussion for “Future experiments” needed to confirm and extend the results regarding the SNP
Author Response
Reviewer 2: The study presents an important finding for an a serious pathogen, Campylobacter jejuni. Particularly, the cysM transcription finding worths publication and is well explained and supported methodologically. I congratulate the authors for way of presentation and details provided and I have only some minor revisions to propose, as follows:
Author response: Thank you for the generous words.
Reviewer 2: Please explain better with one additional phrase in the abstract the “Start site”. Not all readers of a microbiological journal are totally familiar with the concept
Author response: Changes were made to the text as requested.
Reviewer 2: The first sentence of the last paragraph of the introduction is hard to follow, please rephrase
Author response: Changes were made to the text as requested.
Reviewer 2: In discussion, it is referred that “Since the TSS was determined to be slightly upstream of the SNP, it supports the theory that the sequence containing the SNP could play a role in the blocking of full-length transcription.” Why is that? Add a more detailed explanation and define the “slightly”
Author response: Changes were made to the text to clarify our theory that the SNP is a good candidate for playing a role in producing the truncated cysM mRNA transcripts observed in this manuscript and a previously published paper (Gunther 2025).
Reviewer 2: Finally, I suggest to add a small paragraph in the discussion for “Future experiments” needed to confirm and extend the results regarding the SNP
Author response: A future experiments paragraph was added at the end of the discussion section.